# COMPARISON OF PARAGRAM AND GLOVE RESULTS FOR SIMILARITY BENCHMARKS

## ABSTRACT

Distributional Semantics Models(DSM) derive word space from linguistic items in context. Meaning is obtained by defining a distance measure between vectors corresponding to lexical entities. Such vectors present several problems. This work concentrates on quality of word embeddings, improvement of word embedding vectors, applicability of a novel similarity metric used 'on top' of the word embeddings. In this paper we provide comparison between two methods for post process improvements to the baseline DSM vectors. The counter-fitting method which enforces antonymy and synonymy constraints into the Paragram vector space representations recently showed improvement in the vectors' capability for judging semantic similarity. The second method is our novel RESM method applied to GloVe baseline vectors. By applying the hubness reduction method, implementing relational knowledge into the model by retrofitting synonyms and providing a new ranking similarity definition RESM that gives maximum weight to the top vector component values we equal the results for the ESL and TOEFL sets in comparison with our calculations using the Paragram and Paragram + Counter-fitting methods. For SIMLEX-999 gold standard since we cannot use the RESM the results using GloVe and PPDB are significantly worse compared to Paragram. Apparently, counter-fitting corrects hubness. The Paragram or our cosine retrofitting method are state-of-the-art results for the SIMLEX-999 gold standard. They are 0.2 better for SIMLEX-999 than word2vec with sense de-conflation (that was announced to be state-of the-art method for less reliable gold standards). Apparently relational knowledge and counter-fitting is more important for judging semantic similarity than sense determination for words. It is to be mentioned, though that Paragram hyperparameters are fitted to SIMLEX-999 results. The lesson is that many corrections to word embeddings are necessary and methods with more parameters and hyperparameters perform better.

## 1 INTRODUCTION

Distributional language models are frequently used to measure word similarity in natural language. This is a basis for many semantic tasks. The DSM often consists of a set of vectors; each vector corresponds to a character string, which represents a word. Mikolov et al. (2013) and Pennington et al. (2014) implemented the most commonly used word embedding (WE) algorithms. Vector components in language models created by these algorithms are latent. Similarity between words is defined as a function of vectors corresponding to given words. The cosine measure is the most frequently used similarity function, although many other functional forms were attempted. Santus et al. (2016) highlights the fact that the cosine can be outperformed by ranking based functions.

As pointed out by many works, e.g. Hill et al. (2015), evidence suggests that distributional models are far from perfect.

Vector space word representations obtained from purely distributional information of words in large unlabeled corpora are not enough to best the state-of-the-art results in query answering benchmarks, because they suffer from several types of weaknesses:

1. Inadequate definition of similarity,
2. Inability of accounting of senses of words,

3. Appearance of hubness that distorts distances between vectors,

4. Inability of distinguishing from antonyms.

5. In case of retrofitting distortion vector space - loss of information contained in the original vectors

In this paper we use the existing word embedding model but with several post process enhancement techniques. We address three of these problems. In particular, we define a novel similarity measure, dedicated for language models.

Similarity is a function, which is monotonically opposite to distance. As the distance between two given entities gets shorter, entities are more similar. This holds for language models. Similarity between words is equal to similarity between their corresponding vectors. There are various definitions of distance. The most common Euclidean distance is defined as follows:

$$d(p_1, p_2) = \sqrt{\sum_{c \in p}(c_{p_1} - c_{p_2})^2} \tag{1}$$

Similarity based on the Euclidean definition is inverse to the distance:

$$sim(p_1, p_2) = \frac{1}{1 + d(p_1, p_2)} \tag{2}$$

Angular definition of distance is defined with cosine function:

$$d(p_1, p_2) = 1 - cos(p_1, p_2) \tag{3}$$

We define angular similarity as:

$$sim(p_1, p_2) = cos(p_1, p_2) \tag{4}$$

Both Euclidean and Cosine definitions of distance could be looked at as the analysis of vector components. Simple operations, like addition and multiplication work really well in low dimensional spaces. We believe, that applying those metrics in spaces of higher order is not ideal, hence we compare cosine similarity to a measure of distance dedicated for high dimensional spaces.

In this paper we restrict ourselves to three gold standards: TOEFL, ESL and SIMLEX-999. The first two are small but reliably annotated (and therefore confidence in their performance can be assumed 100%). Other used benchmarks suffer from several drawbacks. Both WS-353 Finkelstein et al. (2001) and MEN Bruni et al. (2012) do not measure the ability of models to reflect similarity. Moreover, as pointed out by Hill et al. (2015), for WS-353, RG Rubenstein & Goodenough (1965) and MEN, state-of-the-art models have reached the average performance of a human annotator on these evaluations.

## 2 RELATED WORK

In information retrieval (IR) document similarity to a question phrase can be expressed using ranking. The objective and repeatable comparison of rankings requires a rank similarity measure (e.g. the one introduced by Webber et al. (2010). Santus et al. (2016) introduced another ranking based similarity function called APSyn. In their experiment APSyn outperformed the cosine similarity, reaching 73% of accuracy in the best cases (an improvement of 27% over cosine) on the ESL dataset, and 70% accuracy (an improvement of 10% over cosine) on the TOEFL dataset. In contrast to our work, they use the Positive Pointwise Mutual Information algorithm to create their language model. Despite such large improvement the absolute results were quite poor compared to the state-of-the-art. Our motivation for the RESM method was the NDCG measure that is considered the most appropriate measure for IR evaluation. Levy et al. (2015) investigated Pennington et al. (2014) proposition of using the context vectors in addition to the word vectors as GloVe's output. This led a different interpretation of its effect on the cosine similarity function. The similarity terms which can be divided into two groups: second-order includes a symmetric combination of the first-order and second order similarities of x and y, normalized by a function of their reflective first-order similarities. The best Levy et al. (2015) result for SIMLEX-999 is 0.438; for WordSim-353 similarity and WordSim-353 relatedness their result is a few hundredths worse than that of Paragram Wieting et al. (2015).

A successful avenue to enhance word embeddings was pointed out by Faruqui et al. (2014), using WordNet (Miller & Fellbaum (2007)), and the Paraphrase Database 1.0 (Ganitkevitch et al. (2013)) to provide synonymy relation information to vector optimization equations. They call this process retrofitting, a pattern we adapt to the angular definition of distance, which is more suitable to our case.

We also address hubness reduction. Hubness is related to the phenomenon of concentration of distances - the fact that points get closer at large vector dimensionalities. Hubness is very pronounced for vector dimensions of the order of thousands. To for reduce hubness Dinu & Baroni (2014) used ranking instead of similarity. The method requires the availability of more, unlabeled source space data, in addition to the test instances, and is natural in application to cross-language or cross-modal applications We apply this method of localized centering for hubness reduction (Feldbauer & Flexer (2016)) for the language models.

The Paragram vectors were further improved by Mrksic et al. (2016). This was achieved by the joint retro -fitting and counter-fitting optimization of the following objective function:

$$AP(V') = \sum_{(u,w) \in A} \tau(\delta - d(v'_u, v'_w)))$$ (5)

$$SA(V') = \sum_{(u,w) \in S} \tau(d(v'_u, v'_w) - \gamma))$$ (6)

$$VSP(V, V') = \sum_{i=1}^{N} \sum_{j \in N(i)} \tau(d(v'_u, v'_w) - d(v_u, v_w)))$$ (7)

$$C(V, V') = k_1 AP(V') + k_2 SA(V') + k_3 VSP(V, V')$$ (8)

The values of hyperparameters are the following $\delta = 1$, $\gamma = 0$ and $k_1 = k_2 = k_3$. In the original Paragram + Counter-fitting work $\delta = 1$ corresponds to zero angle between a pair of vectors for synonyms, and $\gamma = 0$ corresponds to 90° angle between a pair of vectors for antonyms, which is not quite consistent with SIMLEX-999 (Hill et al. (2015) average value of cosines for synonym pairs, 0.77, and average value of cosines for antonym pairs, 0.17. However, as we verified, using these theoretically more acceptable values does not improve SIMLEX-999 results.

## 3 METHOD

In our work we define the language model as a set of word representations. Each word is represented by its vector. We refer to a vector corresponding to a word $w_i$ as $v_i$. A complete set of words for a given language is referred to as a vector space model(VSM). We define similarity between words $w_i$ and $w_j$ as a function of vectors $v_i$ and $v_j$.

$$sim(w_i, w_j) = f(v_i, v_j)$$ (9)

We present an algorithm for obtaining optimized similarity measures given a vector space model for word embedding. The algorithm consists of 6 steps:

1. Refine the vector space using the cosine retrofit algorithm
2. Obtain vector space of centroids
3. Obtain vectors for a given pair of words and optionally for given context words
4. Recalculate the vectors using the localized centering method
5. Calculate ranking of vector components for a given pair of words
6. Use the ranking based similarity function to obtain the similarity between a given pair of words.

The cosine retrofit algorithm follows the original Faruqui et al. (2014) approach except that we use the cosine distance and rotations of retrofitted vectors in space. The formulae look different compared to Mrksic et al. (2016) but give equivalent results.

We use all of the methods in together to achieve significant improvement over the baseline method. We present details of the algorithm in the following sections.

### 3.1 BASELINE

The cosine function provides the baseline similarity measure:

$$sim(w_1, w_2) = cos(v_1, v_2) = \frac{v_1 \cdot v_2}{\|v_1\|\|v_2\|} \tag{10}$$

The cosine function achieves a reasonable baseline. It is superior to the Euclidean similarity measure and is used in various works related to word similarity. In our work we use several post-process modifications to the vector space model. We also redefine the similarity measure.

### 3.2 IMPLEMENTING RELATIONAL KNOWLEDGE INTO THE VECTOR SPACE MODEL

Let us define a lexicon of relations (rel) $L$. Each row in the lexicon consists of a word and a set of its related words (synonyms and antonyms).

$$L(w_i) = \{w_j, \beta_j : rel(w_i, w_j), \beta_j = w(rel)\} \tag{11}$$

A basic method of implementing synonym knowledge into the vector space model was previously described in Ganitkevitch et al. (2013). We refer to that method as retrofit. It uses the iterational algorithm of moving the vector towards an average vector of its weighted related words according to the following formula.

$$v_i' = \frac{\alpha_i v_i + \frac{\sum_{w_j \in L_{(}wj)} \beta_j v_j}{\|L(w_j)\|}}{2} \tag{12}$$

In the original formula Faruqui et al. (2014), variables $\alpha$ and $\beta$ allow us to weigh the importance of certain relations. The basic retrofit method moves the vector towards its destination (e.g. shortens the distance between the average synonym to a given vector) using the Euclidean definition of distance. This is not consistent with the cosine distance used for evaluation. Instead we improve Faruqui et al. (2014) idea by performing operations in spherical space, thus preserving the angular definition of distance. Given a basic rotation matrix R for a plane $(i, j)$.

$$R_{(i,j)}(\theta) = \begin{bmatrix} cos\theta & sin\theta \\ -sin\theta & cos\theta \end{bmatrix} \tag{13}$$

For each plain $(p, q)$ we calculate angle $\theta_{(p,q)}(v_i, v_j)$ between word and its related word. We apply a rotation $R_{p,q}(\theta_{(p,q)}(v_i, v_j)\beta_j)$ to that word. Finally, we take an average of a given word vector and average of rotated, related vector.

$$v_i' = \frac{v_i + \frac{\sum_{w_j \in L(w_j)} (\prod_{(p,q)} R_{p,q}(\theta_{p,q}(v_i,v_j)\beta_j))v_i}{\|L(w_j)\|}}{2} \tag{14}$$

We refer to this method as to generalized retrofitting.
The original formula from Faruqui et al. (2014) was obtained by minimizing the following objective function.

$$\Psi(VSM) = \sum_{i=1}^{n} [\alpha_i \|v_i - \hat{v}_i\|^2 + \sum_{w_j \in L(w_i)} \beta_{ij} \|v_i - v_j\|^2] \tag{15}$$

We change the Euclidean definition of distance to cosine in the formula to obtain a new objective function.

$$\Psi'(VSM) = \sum_{i=1}^{n} [(1 - cos(v_i, \hat{v}_i))^2 + \sum_{w_j \in L(w_i)} \beta(i, j)(1 - cos(v_i, v_j))^2] \tag{16}$$

We take first derivative of $\Psi'$ to define a single step change of a vector $v_i$ for each vector component $v_{(i,k)}$.

$$\frac{\delta \Psi'}{\delta v_{(i,k)}} = -2\frac{\delta cos(v_i, \hat{v}_i)}{\delta v_{i,k}} + 2cos(v_i, \hat{v}_i)\frac{\delta cos(v_i, \hat{v}_i)}{\delta v_{i,k}} + \sum_{w_j \in L(w_i)} 2cos(v_i, \hat{v}_i)\frac{\delta cos(v_i, \hat{v}_i)}{\delta v_{i,k}} - 2\beta\frac{\delta cos(v_i, \hat{v}_i)}{\delta v_{i,k}} \tag{17}$$

$$\frac{\delta cos(v_i, v_j)}{\delta v_{i,k}} = \frac{v_{j,k}}{|v_i||v_j|} - cos(v_i, v_j)\frac{v_{i,k}}{|v_i|^2} \tag{18}$$

This method and obtained results are equivalent to the work of Mrksic et al. (2016). We refer to this method as cosine retrofitting.

## 3.3 LOCALIZED CENTERING

We address the problem of hubness in high dimensional spaces with the localized centering approach applied to every vector in the space. The centered values of vectors, centroids, are the average vectors of $k$ nearest neighbors of the given vector $v_i$. We apply a cosine distance measure to calculate the nearest neighbors.

$$c_i = \frac{\sum_{v_j \in k-NN(v_i)} v_i)}{N} \tag{19}$$

In Feldbauer & Flexer (2016), the authors pointed out that skewness of a space has a direct connection to the hubness of vectors. We follow the pattern presented in the Hara et al. (2015) and recalculate the vectors using the following formula.

$$sim(v_i, v_j) = sim'(v_i, v_j) - cos^\gamma(v_i, c_i) \tag{20}$$

We use empirical method to find values of the hyperparameter $gamma$.

### 3.3.1 RANKING BASED SIMILARITY FUNCTION

We propose a component ranking function as the similarity measure. This idea was originally introduced in Santus et al. (2016) who proposed the APSyn ranking function. Let us define the vector $v_i$ as a list of its components.

$$v_i = [f_1, f_2, ..., f_n] \tag{21}$$

We then obtain the ranking $r_i$ by sorting the list in descending order (d in the equation denotes type of ordering), denoting each of the components with its rank on the list.

$$r_i^d = \{f_1 : rank_i^d(f_1), ..., f_n : rank_i^d(f_n)\} \tag{22}$$

APSyn is calculated on the intersection of the N components with the highest score.

$$APSyn(w_i, w_j) = \sum_{f_k \in top(r_i^d) \cap top(r_j^d)} \frac{2}{rank_i(f_k) + rank_j(f_k)} \tag{23}$$

APSyn was originally computed on the PPMI language model, which has unique feature of non-negative vector components. As this feature is not given for every language model, we take into account negative values of the components. We define the negative ranking by sorting the components in ascending order (a in the equation denotes type of ordering).

$$r_i^a = \{f_1 : rank_i^a(f_1), ..., f_n : rank_i^a(f_n)\} \tag{24}$$

As we want our ranking based similarity function to preserve some of the cosine properties, we define score values for each of the components and similarly to the cosine function, multiply the scores

Table 1: a. Example of a question with wrong answer in TOEFL (the correct answer is cushion, our answer is scrape). b. Set of possible questions with the same question word and different correct answers.

|  | Q.word | P1 | P2 | P3 | P4 |
|---|---|---|---|---|---|
| a. | lean | cushion | scrape | grate | refer |

|  | Q.word | P1 | P2 | P3 | P4 |
|---|---|---|---|---|---|
| b. | iron | wood | metal | plastic | timber |
|  | iron | wood | crop | grass | grain |

for each component. As the distribution of component values is Gaussian, we use the exponential function.

$$s_{i,f_k} = e^{-rank_i(f_k)\frac{k}{d}} \tag{25}$$

Parameters $k$ and $d$ correspond respectively to weighting of the score function and the dimensionality of the space. With high k values, the highest ranked component will be the most influential one. The rationale is maximizing information gain. Our measure is similar to infAP and infNDCG measures used in information retrieval Roberts et al. (2017) that give maximum weight to the several top results. In contrast, P@10 assigns equal weight to the top 10 results. Lower $k$ values increase the impact of lower ranked components at the expense of 'long tail' of ranked components. We use the default $k$ value of 10. The score function is identical for both ascending and descending rankings. We address the problem of polysemy with a differential analysis process. Similarity between pair of words is captured by discovering the sense of each word and then comparing two given senses of words. The sense of words is discovered by analysis of their contexts. We define the differential analysis of a component as the sum of all scores for that exact component in each of the context vectors.

$$h_{i,f_k} = \sum_{w_j \in context(w_j)} s_{j,f_k} \tag{26}$$

Finally we define the Ranking based Exponential Similarity Measure (RESM) as follows.

$$RESM^a(w_i, w_j) = \sum_{f_k \in top(r_i^d) \cap top(r_j^d)} \frac{s_{i,f_k}^a s_{j,f_k}^a}{h_{i,f_k}^a} \tag{27}$$

The equation is similar to the cosine function. Both cosine and RESM measures multiply values of each component and sum the obtained results. Contrary to the cosine function, RESM scales with a given context. It should be noted, that we apply differential analysis with a context function $h$. An equation in this form is dedicated for the ESL and TOEFL test sets. The final value is calculated as a sum of the RESM for both types of ordering.

$$RESM(w_i, w_j) = RESM^a(w_i, w_j) + RESM^d(w_i, w_j) \tag{28}$$

## 3.4 IMPLEMENTATION

The algorithm has been implemented in C#. It is publicly available via the repository along with implementation of the evaluation.[1]

## 4 EVALUATION

This work compares the state-of-the-art word embedding methods for three most reliable gold standards: TOEFL, ESL and SIMLEX-999. TOEFL consists of 80 questions, ESL consists of 50 questions. The questions in ESL are significantly harder. Each question in these tests consists of a

---

[1]anonymized repository

Table 2: State of the art results for TOEFL and ESL test sets

| Bullinaria & Levy (2012) Osterlund et al. (2015) | 100.0% | 66.0% |
|---|---|---|
| Jarmasz & Szpakowicz (2012) | 79.7% | 82.0% |
| Lu et al. (2011) | 97.5% | 86.0% |

question word with a set of four candidate answers. It is worth pointing out, that the context given by a set of possible answers often defines the question (a selection of a sense of a word appearing in the question). Answering these tests does not require finding the most similar word out of the whole dictionary but only from multiple choice candidates; therefore TOEFL and ESL are less demanding than SIMLEX-999. A question example in Table 1 highlights the problem. In the first question, all of possible answers are building materials. Wood should be rejected as there is more appropriate answer. In second question, out of possible answers, only wood is a building material which makes it a good candidate for the correct answer. This is a basis for applying a differential analysis in the similarity measure. Table 2. illustrates state of the art results for both test sets. The TOEFL test set was introduced in Landauer & Dumais (1997); the ESL test set was introduced in Turney (2001).

## 4.1 EXPERIMENTAL SETUP

We use the unmodified vector space model trained on 840 billion words from Common Crawl data with the GloVe algorithm introduced in Pennington et al. (2014). The model consists of 2.2 million unique vectors; Each vector consists of 300 components. The model can be obtained via the GloVe authors website. We also use the GloVe refinement enriched with PPDB 1.0 relations called Paragram Wieting et al. (2015) (there is an extension of this paper at https://arxiv.org/pdf/1506.03487.pdf with more details included). We run several experiments, for which settings are as follows: In the evaluation skewness $\gamma = 9$ and $k = 10$. All of the possible answers are taken as context words for the differential analysis. In our runs we use all of the described methods separately and conjunctively. We refer to the methods in the following way. We denote the localized centering method for hubness reduction as HR. We use a Paraphrase Database lexicon introduced in Ganitkevitch et al. (2013) for the retrofitting. We denote the generalized retrofitting as RETRO and the cosine retrofitting as cosRETRO.

## 4.2 HEURISTIC IMPROVEMENT

Although the hubness reduction method does not increase the number of correct answers for the ESL test set, we noticed that the average rank of the correct answer goes down from 1.32 to 1.24. That is a significant improvement. To obtain better results we combined the results with and without localized centering. The heuristic method chooses the answer with the best average rank for both sets. By applying that method we obtained two additional correct answers.

## 4.3 SENSE RECOGNITION

Semantic distributional methods operate on words or characters. A vector position for a word is an average of positions of the word senses. The is a major deficiency, particularly if a word with a given sense is statistically rare. In Table 4 of Mrksic et al. (2016) a pair: dumb dense appears amount the highest-error SIMLEX-999 word pairs using Paragram vectors (before counter-fitting). Counter-fitting does not improve this result.

In Table 3 we also show the results sense recognition systems. The approach of Pilehvar & Navigli (2015) is purely based on the semantic network of WordNet and does not use any pre-trained word embeddings. In their approach, similarly to the VSM representation of a linguistic item, the weight associated with a dimension in a semantic signature denotes the relevance or importance of that dimension for the linguistic item. In Pilehvar & Collier this method was improved using the Faruqui et al. (2014) like formula, with senses in place of synonyms or antonyms. In this DECONF method the word embeddings were used following Mikolov et al. (2013).They obtained state-of-the-art results for MEN-3K, and RG-65 (here Spearman correlation 0.896 by far exceeds the inter-annotator

Table 3: Accuracy of various methods on TOEFL and ESL test sets. *cosRETRO+ is trained on all relations from PPDB(Equivalence, Exclusion, ForwardEntilement, ReverseEntilement, Other-Relation, Independent), while cosRETRO is trained on Equivalence from PPDB as synonyms and Exclusion from PPDB with Antonyms from WordNet as antonyms).

| Method | TOEFL | ESL | SIMLEX-999 |
|---|---|---|---|
| Cosine | 88.75% | 60.00% | 0.435 |
| HR + Cosine | 91.25% | 66.00% | 0.438 |
| RETRO + Cosine | 95.00% | 62.00% | 0.435 |
| HR + RETRO + Cosine | 96.25% | 74.00% | 0.438 |
| APSyn | 80.00% | 60.00% | 0.338 |
| RETRO + APSyn | 97.50% | 70% | Not Applicable |
| RESM | 90.00% | 76.00% | Not Applicable |
| RETRO + RESM | 96.25% | 80.00% | Not Applicable |
| HR + RETRO + RESM | 97.50% | 80.00% | Not Applicable |
| RETRO + RESM + heuristic | 97.50% | 84.00% | Not Applicable |
| Paragram | 97.50% | 84.00% | 0.688 |
| Paragram + HR | 97.50% | 84.00% | 0.692 |
| Paragram + Counter-fitting | 97.50% | 82.00% | 0.736 |
| Paragram + cosRETRO* | 97.50% | 82.00% | 0.724 |
| Paragram + cosRETRO+* | 97.50% | 84.00% | 0.716 |
| Pilehvar and Navigli | | | 0.436 |
| DECONF | | | 0.517 |

confidence). However, for SIMLEX-999 the multi-sense results are 0.3 worse (Pilehvar & Navigli (2015)) and 0.2 worse in Pilehvar & Collier compared to the Paragram based results.

We improved the accuracy results by 8.75% and 24% for TOEFL and ESL test sets respectively. We observe the largest improvement of accuracy by applying the localized centering method for TOEFL test set. Testing on the ESL question set seems to give the best results by changing the similarity measure from cosine to RESM. Thus each step of the algorithm improves the results. The significance of the improvement varies, depending on the test set. A complete set of measured accuracies is presented in Table 3. The results for the APSyn ranking method are obtained using the Glove vector, not using PPMI as in Santus et al. (2016).

The performance of Paragram with counter-fitting for SIMLEX-999 is better than inter-annotator agreement, measured by average pairwise Spearman $\rho$ correlation, which varies between 0.614 to 0.792 depending on concept types in SIMLEX-999.

There are many results that are significantly worse than these in Table 3. For example, in Kiela et al. (2015) the accuracy for the TOEFL standard is 88.75% and for SIMLEX-999 the Spearman $\rho$ is 0.53.

## 5 CONCLUSIONS

This work compares the state-of-the-art word embedding methods for three most reliable gold standards: TOEFL, ESL and SIMLEX-999. For TOEFL and ESL the GloVe, PPDB baseline with retrofitting, our novel RESM similarity measure and hubness reduction we are able to equal the Paragram results. For SIMLEX-999 Paragram with Counter-fitting results are clearly better than the Glove based methods using the PPDB 1.0. However, we propose the cosine retrofitting that basically achieves the Paragram with Counter-fitting results. The Paragram with Counter-fitting method contains several hyperparameters which is one source of its success. Its effects can be seen in Table 10 at https://arxiv.org/pdf/1506.03487.pdf. The Spearman $\rho$ values for SIMLEX-999 are 0.667 for Paragram300 fitted to WS353, and 0.685 for Paragram300 fitted to SIMLEX-999. The difference is even larger for WS353. Then the Spearman $\rho$ values for WS-353 are 0. 769 for Paragram300 fitted toWS353, and 0.720 for Paragram300 fitted to SIMLEX-999. Still the best word embedding based methods are not able to achieve the performance of other dedicated methods for TOEFL and

ESL. The work of Lu et al. (2011) employed 2 fitting constants (and it is not clear that they were the same for all questions) for answering the TOEFL test where only 50 questions are used. Techniques introduced in the paper are lightweight and easy to implement, yet they provide a significant performance boost to the language model. Since the single word embedding is a basic element of any semantic task one can expect a significant improvement of results for these tasks. In particular, SemEval-2017 International Workshop on Semantic Evaluation run (among others) the following tasks(se2):

1. Task 1: Semantic Textual Similarity
2. Task 2: Multilingual and Cross-lingual Semantic Word Similarity
3. Task 3: Community Question Answering

in the category Semantic comparison for words and texts. Another immediate application would be information retrieval (IR). Expanding queries by adding potentially relevant terms is a common practice in improving relevance in IR systems. There are many methods of query expansion. Relevance feedback takes the documents on top of a ranking list and adds terms appearing in these document to a new query. In this work we use the idea to add synonyms and other similar terms to query terms before the pseudo- relevance feedback. This type of expansion can be divided into two categories. The first category involves the use of ontologies or lexicons (relational knowledge). The second category is word embedding (WE). Here closed words for expansion have to be very precise, otherwise a query drift may occur, and precision and accuracy of retrieval may deteriorate.
There are several avenues to further improve the similarity results.

1. Use the multi-language version of the methods (e.g. Recski et al. (2016))
2. Use PPDB 2.0 to design the Paragram vectors Pavlick et al. (2015)
3. Apply the multi-sense methods (knowledge graps) with state-of-the-art relational enriched vectors
4. Recalibrate annotation results using state-of-the-art results.

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
