# OpenReview forum: "Comparison of Paragram and GloVe Results for Similarity Benchmarks"
_ICLR.cc/2018/Conference — Reject_

### Official Review · AnonReviewer2 · 2017-11-27
**Main point of paper is unclear and unproven**

**Rating:** 2
**Confidence:** 4

**Review:**

The paper suggests taking GloVe word vectors, adjust them, and then use a non-Euclidean similarity function between them. The idea is tested on very small data sets (80 and 50 examples, respectively). The proposed techniques are a combination of previously published steps, and the new algorithm fails to reach state-of-the-art on the tiny data sets.

It isn't clear what the authors are trying to prove, nor whether they have successfully proven what they are trying to prove. Is the point that GloVe is a bad algorithm? That these steps are general? If the latter, then the experimental results are far weaker than what I would find convincing. Why not try on multiple different word embeddings? What happens if you start with random vectors? What happens when you try a bigger data set or a more complex problem?

---

### Official Review · AnonReviewer1 · 2017-11-28

**Rating:** 4
**Confidence:** 5

**Review:**

This paper proposes a ranking-based similarity metric for distributional semantic models. The main idea is to learn "baseline" word embeddings, retrofitting those and applying localized centering, to then calculate similarity using a measure called "Ranking-based Exponential Similarity Measure" (RESM), which is based on the recently proposed APSyn measure.

I think the work has several important issues:

1. The work is very light on references. There is a lot of previous work on evaluating similarity in word embeddings (e.g. Hill et al, a lot of the papers in RepEval workshops, etc.); specialization for similarity of word embeddings (e.g. Kiela et al., Mrksic et al., and many others); multi-sense embeddings (e.g. from Navigli's group); and the hubness problem (e.g. Dinu et al.). For the localized centering approach, Hara et al.'s introduced that method. None of this work is cited, which I find inexcusable.

2. The evaluation is limited, in that the standard evaluations (e.g. SimLex would be a good one to add, as well as many others, please refer to the literature) are not used and there is no comparison to previous work. The results are also presented in a confusing way, with the current state of the art results separate from the main results of the paper. It is unclear what exactly helps, in which case, and why.

3. There are technical issues with what is presented, with some seemingly factual errors. For example, "In this case we could apply the inversion, however it is much more convinient [sic] to take the negative of distance. Number 1 in the equation stands for the normalizing, hence the similarity is defined as follows" - the 1 does not stand for normalizing, that is the way to invert the cosine distance (put differently, cosine distance is 1-cosine similarity, which is a metric in Euclidean space due to the properties of the dot product). Another example, "are obtained using the GloVe vector, not using PPMI" - there are close relationships between what GloVe learns and PPMI, which the authors seem unaware of (see e.g. the GloVe paper and Omer Levy's work).

4. Then there is the additional question, why should we care? The paper does not really motivate why it is important to score well on these tests: these kinds of tests are often used as ways to measure the quality of word embeddings, but in this case the main contribution is the similarity metric used *on top* of the word embeddings. In other words, what is supposed to be the take-away, and why should we care?

As such, I do not recommend it for acceptance - it needs significant work before it can be accepted at a conference.

Minor points:
- Typo in Eq 10
- Typo on page 6 (/cite instead of \cite)

---

### Official Review · AnonReviewer3 · 2017-11-30
**A set of retrofitting methods for measuring lexical similarity**

**Rating:** 3
**Confidence:** 4

**Review:**

I hate to say that the current version of this paper is not ready, as it is poorly written. The authors present some observations of the weaknesses of the existing vector space models and list a 6-step approach for refining existing word vectors (GloVe in this work), and test the refined vectors on 80 TOEFL questions and 50 ESL questions. In addition to the incoherent presentation, the proposed method lacks proper justification. Given the small size of the datasets, it is also unclear how generalizable the approach is.

Pros:
  1. Experimental study on retrofitting existing word vectors for ESL and TOEFL lexical similarity datasets

Cons:  1. The paper is poorly written and the proposed methods are not well justified.
  2. Results on tiny datasets

---

### Author Response · Authors · 2018-01-05
**The comments for all reviewers**

The original paper was very significantly changed, expanded (8,5 instead 6 pages). This work concentrates on  quality of word embeddings, improvement of word embedding vectors, applicability of a novel similarity metric used ‘on top’ of the word embeddings. The comparison of our cosine retrofitting to Paragram + Counterfitting  for SIMLEX -999; and our  RESM + cosine retrofitting to Paragram was done.
In particular this revision provides the following:

1.	Improves the clarity of the original version by almost twice as many experimental details; also in the area of what is state-of-the-art and what is not (using reliable gold standards, and concentrating on absolute results rather than on result changes often caused by a single effect).
2. Removes a major deficiency of the original paper by including and addressing the Paragram and Paragram + Counter-fitting  methods’ results.
3. Adds all references that were considered necessary by reviewers. It is not that we were not aware of most of them. Notice that there was a one page limit on references. It seems we were one of a very few to obey this rule.
4. In addition to TOEFL and ESL we included the SIMLEX-999 standard. We consider them the only reliably annotated sets at the moment for two reasons already mentioned by [1].
5. The main results in Table 3 were augmented by Paragram, and Paragram + Counter-fitting  methods and the multi-sense aware methods (Pilehvar and Navigli) .
There are many important conclusions reached in this paper: mostly many corrections to word embeddings are necessary for  state-of-the-art results, and methods with more parameters and hyperparameters perform better.


[1] Hill,  Reichart, and  Korhonen. Simlex-999: Evaluating semantic models with (genuine)
similarity estimation. Computational Linguistics, , 2015.

---

### Decision · Program_Chairs · 2018-01-29
**ICLR 2018 Conference Acceptance Decision**

**Decision:**

Reject

**Comment:**

This paper proposes a method for refining distributional semantic representation at the lexical level. The reviews are fairly unanimous in that they found both the initial version of the paper, which was deemed quite rushed, and the substantial revision unworthy of publication in their current state. The weakness of both the motivation and the experimental results, as well as the lack of a clear hypothesis being tested, or of an explanation as to why the proposed method should work, indicates that this work needs revision and further evaluation beyond what is possible for this conference. I unfortunately must recommend rejection.